# Emotional Distress Caused by the Measures Taken in Assisted Reproductive Treatments during the COVID-19 Confinement in Spain

**DOI:** 10.3390/jcm12227069

**Published:** 2023-11-13

**Authors:** Marta Correa Rancel, Elena Sosa Comino, Fatima Leon-Larios, Yaiza Suárez Hernández, Janet Carballo Lorenzo, Diego Gomez-Baya, Delia Baez Quintana

**Affiliations:** 1Human Reproduction Unit, Gynecology Service, University Hospital of the Canary Islands, 38320 Tenerife, Spain; tenerife1833@gmail.com (M.C.R.); elena_sc@hotmail.com (E.S.C.); yaizash@gmail.com (Y.S.H.); janetcarballo76@gmail.com (J.C.L.); drbaez@ull.edu.es (D.B.Q.); 2Departamento de Obstetricia-Ginecología, Pediatría, Preventiva, Medicina Legal y Forense, Microbiología, Parasitología, Universidad de la Laguna, 38200 Santa Cruz de Tenerife, Spain; 3Nursing Department, University of Seville, 41004 Sevilla, Spain; 4Department of Social, Developmental and Educational Psychology, Universidad de Huelva, 21007 Huelva, Spain; diego.gomez@dpee.uhu.es

**Keywords:** COVID-19, fertility, anxiety, infertility, confinement, lockdown

## Abstract

During the pandemic, assisted reproductive treatments suffered from major disruptions in their terms due to the restrictions imposed. The objective of this study is to evaluate the level of anxiety of women whose treatments were either suspended or delayed. Methods: Descriptive cross-sectional study conducted between April and May 2020. The State-Trait Anxiety Inventory was applied by telephone in a Spanish adapted version. The research also included social, personal, and work aspects which may be involved in the challenging situation. Results: A total of 115 patients participated in the study (73.7%). Women showed a mean in trait anxiety of 17.79 (SD = 8.80) and a mean in state anxiety of 19.95 (SD = 9.08). Neither the type of treatment nor the time of infertility were predictors of trait anxiety or state anxiety. Greater age pressure and more worry were associated to greater trait and state anxiety (*p* < 0.001). The most common emotional reactions to discontinuation of fertility treatments were sadness and anxiety. Conclusions: Discontinuation of fertility treatments due to confinement restrictions had a negative impact on the mental health of women who were following a process of assisted reproduction treatment, increasing their levels of emotional distress and anxiety.

## 1. Introduction

Nowadays, in Spain, assisted reproductive treatments affect 9% of the total annual number of pregnancies [1]; the numbers are increasing year after year. This difficulty in conceiving generates a considerable amount of stress, frustration, and anxiety. It is estimated that a great number of patients at assisted reproductive technology services show clinical symptoms of emotional affectation [2,3,4] which may have a negative impact on the mental health of couples. This situation makes them more vulnerable to depression and anxiety. Hence, psychological support during the process is important [5,6].

The state of emergency due to the pandemic caused by the SARS-CoV-2 virus temporarily affected the management of the activities of the Assisted Reproduction Units in Spain, as these types of services were not considered a health priority. Scientific societies related to these objectives—European Society of Human Reproduction and Embryology (ESHRE), Sociedad Española de Fertilidad (SEF), Asociación para el estudio de la Biología de la Reproducción, (ASEBIR)—recommended women to postpone the start of new treatments until the end of the state of alarm, finish the ongoing cycles and not transfer but have their embryos frozen. Fertility preservation cycles were only initiated in cancer patients, before gonadotoxic treatments [7]. All this had an impact on patients who were waiting for their treatment and could not afford to delay it any longer. This situation caused an increase in the risk of psychological distress [8].

There were no recent precedents for a pandemic of this magnitude. However, there is evidence that emotional distress had a strong influence in this situation, both on health professionals and the general population [9]. There was a generalized fear of contagion, and feelings of anxiety and depression were common [10,11]. Fertility treatments normally require multiple face-to-face appointments in which patients interact with a multidisciplinary team for the necessary tests, a routine which may spread contagion. The European Society for Human Reproduction and Embryology (ESHRE), as well as other societies, recommended to suspend new treatments and non-urgent surgical procedures. 

The aim of this study was to assess the level of anxiety of patients whose assisted reproduction treatments were either suspended or postponed until the end of confinement caused by the COVID-19 pandemic, as well as to evaluate the different sociodemographic variables associated to the emotional management of this situation.

## 2. Materials and Methods

A descriptive cross-sectional study was conducted during April and May 2020. Participant recruitment followed a consecutive random sampling procedure conducted during the confinement months among women included in the assisted reproduction program at the Canarias university hospital (Santa Cruz de Tenerife, Spain), a tertiary hospital belonging to the Public Health System. All women included in the program were invited to participate in the study. The inclusion criteria were women aged 27–42 years who were receiving fertility treatments: (1) In-vitro fertilization- intracytoplasmic sperm injection (IVF-ICSI) patients who already started their treatment with analogues of GnRH (initial ovulation induction phase) or contraceptives, or those patients who had their medication but were waiting for their periods to start the treatment; (2) patients in the procedure cycle of frozen embryo transfer (FET); (3) patients undergoing the treatment of ovarian stimulation for artificial insemination (AI). Exclusion criteria were participation non-acceptance and not having initiated the reproductive process being currently under study.

### 2.1. Instruments

Psychological tests evaluate multidimensional traits and subfactors, such as personality, anxiety, or depression. Each patient was administered a validated diagnostic test by telephone called the STAI (State-Trait Anxiety Inventory. Spanish adaptation: Spielberg, Gorsuch, Lushere (1983)) [12]. The aim of the STAI questionnaire is to evaluate two concepts: anxiety as a state (emotional, transitory condition) and anxiety as a trait (stable tendency to experience and report negative emotions such as fear or worry). The time frame of reference in the case of anxiety as a state is “Right now, at this moment”, and in the case of anxiety as a trait is “In general, most of the time”. Each of these concepts are measured through 20 items in a 4-Point Likert Scale response according to intensity (0 = almost never/never; 1 = something/sometimes; 2 = very much/very often; 3 = a lot/almost always). Total punctuation in each subscale ranges from 0 to 60 points. A STAI score of 34–36 was considered normal. This is a validated questionnaire, widely used in the literature to evaluate trait anxiety and state anxiety levels in both general and clinical populations, being also one of the most used by the Spanish psychologists Muñiz and Fernández-Hermida [13]. The state anxiety scale obtained a Reliability Index rating of α = 0.86 and the trait anxiety scale obtained that of α = 0.78, thus assuring the internal consistency reliability obtained by the sample.

Then, a structured survey developed by the Human Reproduction Unit team was administered to assess other social, occupational, and personal aspects that could be affecting the way patients were facing the situation. 

### 2.2. Ethical Considerations

Patients were interviewed on the phone after obtaining their verbal consent to participate and providing the information regarding objectives and implications of collaborating in the study. The interviews were conducted by a female collaborator who was not involved in the interviewees’ health care. Institutional Review Board (IRB) approval was obtained (CHUC_2020_36) and followed the principles established in the Declaration of Helsinki. 

### 2.3. Data Analysis Design

First, descriptive statistics were presented, by showing percentage distribution in categorical variables (i.e., type of treatment, sterility time, and psychological variables), and means and standard deviations in both trait and state anxiety. Second, differences in psychological variables by treatment type and sterility time were calculated by performing Χ^2^ tests. Likewise, differences in anxiety by treatment type, sterility time, and psychological variables were calculated by means of variance analyses. For these bivariate analyses, φ and η^2^_p_ were presented, respectively, to provide information about effect size. Third, two hierarchical regression analyses were conducted to separately explain both trait and state anxiety based on treatment type, sterility time, and psychological variables. Standardized coefficients and R^2^ were presented. These analyses were performed using SPSS 21. 

## 3. Results

A total of 156 assisted reproduction patients were invited to participate in this study, where 115 finally agreed to reply to the questionnaire (73.7%). The age of the patients ranged from 27 to 42 years, with an average of 34.84 (SD = 4.13). Participants received one of three different treatments: IVF-ICSI (*n* = 56, 48.7%), FET (*n* = 44, 38.3%), or AI (*n* = 15, 13%). Regarding sterility time, most of the patients were sterile for less than five years (83.5%), while some participants reported more than five years (15.5% 5–10 years, 1% more than 11 years) of sterility. Table 1 presents the frequency distribution of the variables regarding the psychological responses to treatment during the pandemic. When participants were told that they had to postpone the treatment, most of them felt sad (54.1%) or frustrated (32.7). However, most of them responded to it with acceptance (65.2%) or tranquility (26.1%). Concerning age pressure, 32.2% of participants felt moderate pressure and 26.1% felt a lot or quite a lot of pressure. Furthermore, 37.7% of participants felt moderate worry about pregnancy during the pandemic, and 18.4% felt even higher worry. When women were asked about coping against a setback, most of them reported that they would take the good advice of the professionals (75.7%). Finally, concerning anxiety levels, with scores in trait and state anxiety ranging from 0 to 60, results showed a mean in trait anxiety of 17.79 (SD = 8.80) and a mean in state anxiety of 19.95 (SD = 9.08). 

In general, the whole sample shows relatively low values of both state anxiety and trait anxiety, none of them reaching the 0.5 quantile median. However, the high standard deviation in both dimensions is noteworthy. Analyzing the anxiety level per treatment type, a significant interaction is found between the type of anxiety and the type of treatment (F(2.112) = 9.02; *p* < 0.001; η_p_^2^ = 0.139). Post hoc tests reveal that in both measurements, FET patients show significantly lower levels with respect to IVF/ICSI or AI patients (*p* < 0.001), besides being the only group of patients who do not show significant differences between trait and state anxiety of the studied period, as illustrated by Table 2.

Figure 1 and Figure 2 show percentage distribution in age pressure and means in trait anxiety by type of treatment. More age pressure is reported by participants with IVF-ICSI treatment, while less pressure is felt by those with AI treatment, Χ^2^ (8) = 19.91, *p* = 0.011, φ = 0.416. Moreover, more trait anxiety is presented by participants in FET and IVF-ICSI treatments, F(2, 112) = 6.30, *p* = 0.003, η^2^_p_ = 0.101. No differences are detected in state anxiety, F(2, 111) = 1.85, *p* = 0.162, nor with regard to sterility time.

Figure 3 and Figure 4 present the means in trait and state anxiety by the age pressure and the worry about living a pregnancy during the pandemic. The results indicated that participants who reported quite a lot of age pressure presented more trait anxiety, F(4, 110) = 3.01, *p* = 0.021, η^2^_p_ = 0.099, and more state anxiety, F(4, 109) = 4.58, *p* = 0.002, η^2^_p_ = 0.144. Furthermore, those participants who felt a lot of worry about living a pregnancy during the pandemic showed more trait anxiety, F(4, 109) = 4.09, *p* = 0.004, η^2^_p_ = 0.130, and more state anxiety, F(4, 108) = 6.37, *p* < 0.001, η^2^_p_ = 0.191. No differences in anxiety were observed in the other psychological variables. 

Furthermore, Figure 5 and Figure 6 show differences in trait and state anxiety by the type of treatment and the level of age pressure. In trait anxiety, significant differences were observed in the groups of IVF-ICSI, F(4, 51) = 3.76, *p* = 0.009, η^2^_p_ = 0.228, and AI treatments, F(4, 10) = 4.35, *p* = 0.027, η^2^_p_ = 0.635, so greater trait anxiety was found in participants with quite a lot of age pressure. Moreover, with regard to state anxiety, the differences only reached significance in the group of treatment by IVF-ICSI, F(4, 51) = 5.32, *p* = 0.001.

Figure 7 and Figure 8 present the differences in trait and state anxiety by the type of treatment and the level of worry about pregnancy during the pandemic. Differences in trait anxiety were only detected in the group of IVF-ICSI treatment, F(4, 51) = 4.53, *p* = 0.003, η^2^_p_ = 0.262, with higher anxiety in participants with higher worry. Moreover, differences in state anxiety by the levels of worry were found in the groups of IVF-ICSI treatment, F(4, 51) = 4.70, *p* = 0.003, η^2^_p_ = 0.269, and FET, F(4, 37) = 2.69, *p* = 0.046, η^2^_p_ = 0.225.

### Multivariate Analyses

Table 3 describes the results of two hierarchical regression models to explain state and trait anxiety based on treatment type, sterility time and psychological variables. Results indicated that neither treatment nor sterility time were significant predictors of anxiety (both trait and state) when psychological variables were included in the regression equation. Thus, only age pressure and worry about pregnancy during pandemic had positive effects on trait and state anxiety, reaching explained variance over 20%. In line with previous results, more age pressure and more worry were associated to greater trait and state anxiety.

## 4. Discussion

The aim of this study was to analyze the emotional distress caused by the health measures adopted during confinement in women who were waiting for their assisted reproduction treatments when they were cancelled, as well as to evaluate the associated variables. 

The mean of the STAI questionnaire in our study was lower than those of previous researchers [6]. Low levels of both trait and state anxiety may be related with the fact that the incidence of the pandemic in the Canary Islands was among the lowest in the country. However, similar values were observed in women when the type of fertility treatment is IVF-ICSI. Both state and trait anxiety were affected in the being pregnant during the COVID pandemic dimension [6].

With respect to fertility treatments, women of the IVF group showed more anxiety than those in the artificial insemination treatment group, which could be explained by age pressure and the ovarian reserve [6,14,15]. In line with this, they showed more trait anxiety, though no differences were observed with respect to state anxiety or related to the time of infertility. Therefore, neither the type of treatment nor the time of infertility were predictors of span of trait and state anxiety, something that other studies highlighted as influencing factors [6,16].

Those women showing a greater concern about becoming pregnant during the pandemic developed a higher level of state–trait anxiety. There were no differences in the rest of psychological variables, in line with the results of the study by Haham et al. [17]. Whereas in this study, the most common emotional reactions to the suspension of treatment were sadness and anxiety, in the study by Qu et al. [11] it was fear, and in our case they were sadness and frustration. 

To the emotional distress involved in fertility treatments it is necessary to add the one experienced during the containment measures introduced on the COVID-19 pandemic [18]. According to the study by Turocy et al., the COVID-19 pandemic affected 85% of participants who were moderately to extremely distressed by their fertility cycle cancellation [19] regardless of the type of treatment. Nearly a quarter of participants found the delay to be as distressing as loss of a child. Only a third agreed with treatment cancellation and more than half would prefer to continue despite the risk posed by the pandemic. Notwithstanding, most of them accepted the situation. These findings are in line with the results found in our study and other research [6,16,17,20]. We agree with the information available in other studies, which point out that interruptions or delays of fertility treatments may cause a negative impact in the anxiety experienced by women who saw their maternity compromised [6,11,14,21,22]. For this very reason, having the embryos frozen could be a safe haven asset for situations like the ones experienced during the pandemic. 

The advice and information received by the professionals involved in assisted reproductive therapies are essential to cope with the situation of stress generated [16]. This situation was aggravated in the confinement during the pandemic, where the need for follow-up and information provided to patients increased significantly. As illustrated by the participants in this study, 7 out of 10 reported relying on healthcare professionals to manage the situation. The information provided by healthcare professionals is of great value in fertility treatments, and it was especially relevant in the pandemic situation experienced by the participants in this study. This was corroborated by other studies such as the one by Lawson et al., which indicated that supplemental education increased the acceptance of the recommendations, though the stress caused by the situations experienced was not reduced [16]. As in previous studies, the use of telemedicine seems to help to reduce both stress and anxiety experienced [23]. Therefore, in situations of pandemic, it may be a good approach to keep providing the best care services [24]. 

Among the limitations of this study, we could identify the fact that it lacks the participants’ anxiety baseline condition records which may allow identification of their initial situation before pandemic restrictions. In addition, it is not possible to identify whether the values are exclusively associated to the fertility treatments or to all the situations experienced in relation to the pandemic, as other studies pointed out [25]. There are no records of patient anxiety or mental health disorders prior to the pandemic. On the other hand, the questionnaires were administered by telephone due to the restrictions imposed by the pandemic situation on attending appointments.

## 5. Conclusions

Suspension of fertility treatments due to confinement measures had a negative impact on the mental health of women who were in the process of receiving assisted reproduction treatment, increasing their levels of emotional distress and anxiety. Age pressure and pregnancy during the pandemic influenced the levels of state–trait anxiety experienced, and therefore it is necessary to establish certain strategies which allow the retention of the greatest possible number of services associated to the assisted reproductive techniques in order to reduce the impact on the mental health of patients of fertility treatments, as well as planning a specific network of psychological support in situations where services are either suspended or delayed.

## Figures and Tables

**Figure 1 jcm-12-07069-f001:**
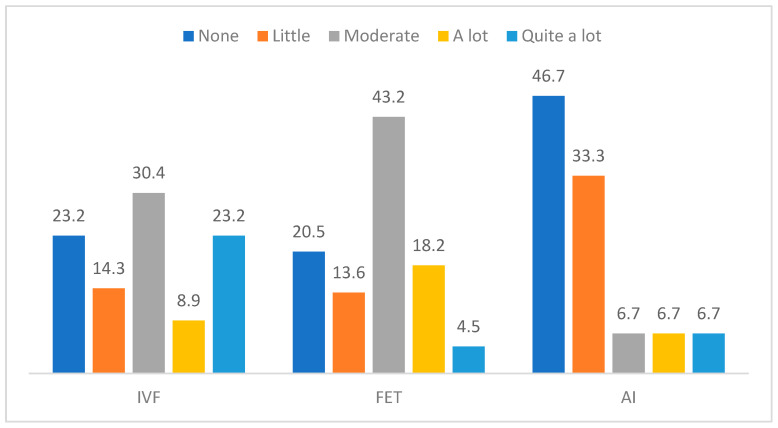
Differences in age pressure by the type of treatment.

**Figure 2 jcm-12-07069-f002:**
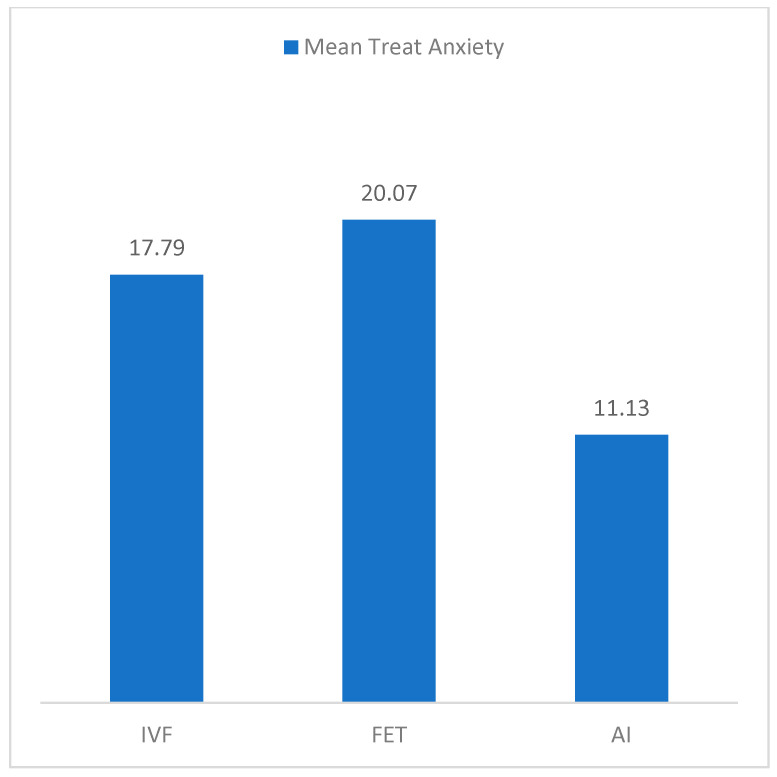
Differences in age pressure and trait anxiety by the type of treatment.

**Figure 3 jcm-12-07069-f003:**
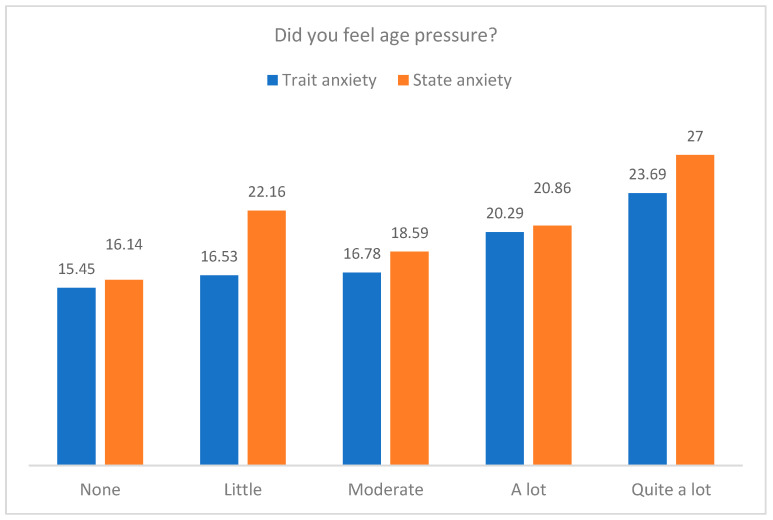
Means in trait and state anxiety by the levels in age pressure.

**Figure 4 jcm-12-07069-f004:**
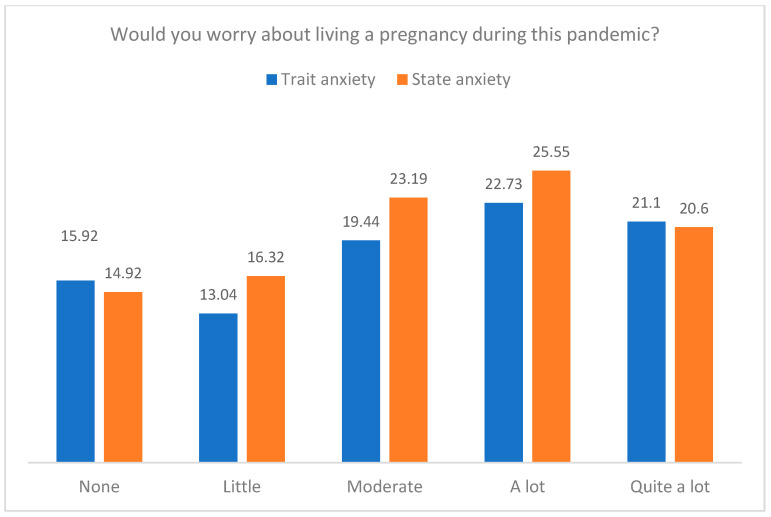
Means in trait and state anxiety by the levels in worry about living a pregnancy during this pandemic.

**Figure 5 jcm-12-07069-f005:**
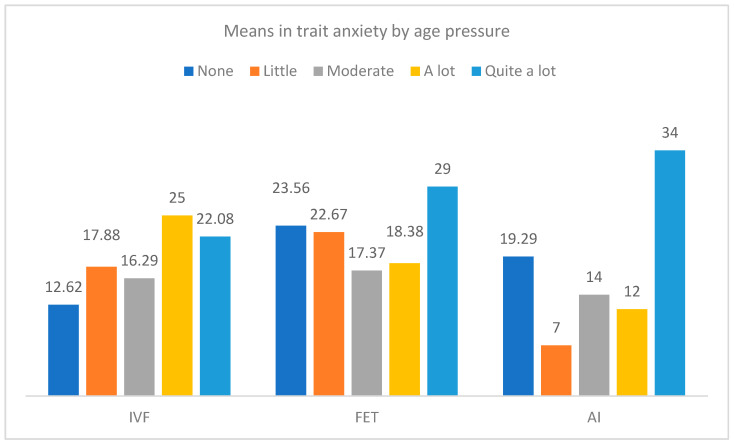
Means in trait anxiety by the type of treatment and the level of age pressure.

**Figure 6 jcm-12-07069-f006:**
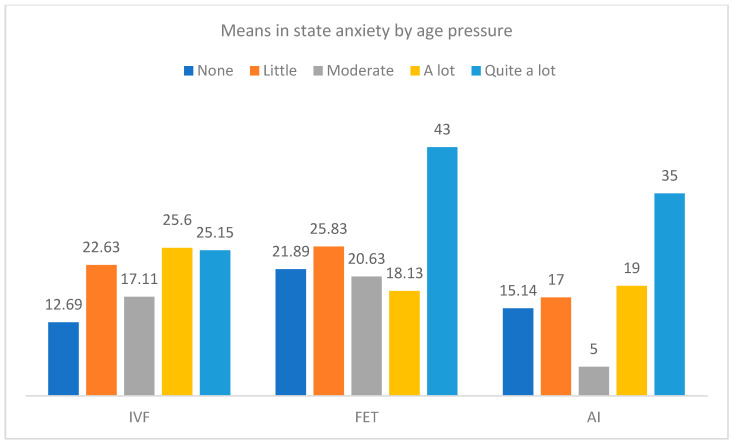
Means in state anxiety by the type of treatment and the level of age pressure.

**Figure 7 jcm-12-07069-f007:**
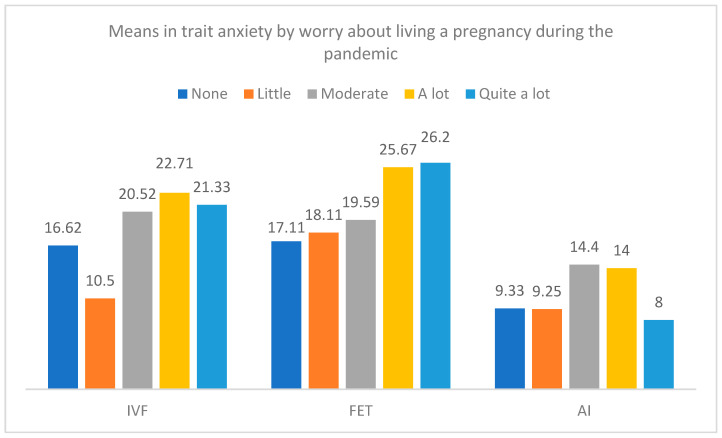
Means in trait anxiety by the type of treatment and the level of worry about living a pregnancy during the pandemic.

**Figure 8 jcm-12-07069-f008:**
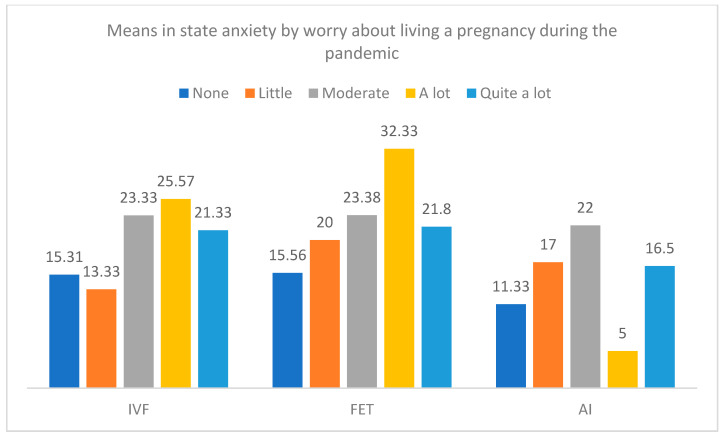
Means in state anxiety by the type of treatment and the level of worry about living a pregnancy during the pandemic.

**Table 1 jcm-12-07069-t001:** Percentage distribution of the variables of psychological responses to treatment during pandemic.

	Anguish	Frustration	Sadness	Loneliness	
What did you feel when you were told that you had to postpone your treatment?	10.2	32.7	54.1	3.1	
	Relief	Tranquility	Relaxation	Acceptance	
How did you experience it?	5.2	26.1	3.5	65.2	
	None	Little	Moderate	A lot	Quite a lot
Did you feel age pressure?	25.2	16.5	32.2	12.2	13.9
Would you worry about pregnancy during this pandemic?	21.9	21.9	37.7	9.6	8.8
	No matter what happens, I will continue.	I wonder if it is convenient to continue.	I take the good advice of professionals.	I abandon because it is a sign that I should not continue.	
When a setback arises regarding your goal…	13	11.3	75.7	0	

**Table 2 jcm-12-07069-t002:** Descriptive values of STAI dimensions, Mean ± Standard deviation.

Type of Treatment	*n*	Trait Anxiety Quantile		State Anxiety Quantile	
IVF-ICSI	56	41.27	±25.54	61.39	±18.53
FET	44	16.93	±17.05	19.91	±15.71
AI	15	37.73	±28.26	55.33	±11.10
Total	115	31.5	±25.62	44.73	±25.75

**Table 3 jcm-12-07069-t003:** Hierarchical regression models to explain state and trait anxiety based on treatment type, sterility time and psychological variables.

	Dependent Variable: State Anxiety				Dependent Variable: Trait Anxiety			
	F	R^2^	t	β	F	R^2^	t	β
	4.51 ***	0.24			4.07 **	0.21		
Treatment			0.98	0.10			0.71	0.97
Sterility time			−1.03	−0.11			−0.67	−0.07
What did you feel when you were told that you had to postpone your treatment?			0.72	0.07			0.84	0.09
How did you experience it?			−0.14	−0.01			−1.38	−0.14
Did you feel age pressure?			2.84	0.31 **			2.59	0.28 *
Would you worry about pregnancy within this pandemic?			3.04	0.32 **			3.08	0.33 **
When a setback arises in regard to your goal…			1.91	0.20			1.48	0.16

* *p* < 0.05; ** *p* < 0.01; *** *p* < 0.001.

## Data Availability

The data presented in this study are available on request from the corresponding author.

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
