# Peer review of "Emotional Distress Caused by the Measures Taken in Assisted Reproductive Treatments during the COVID-19 Confinement in Spain"

_jcm, 2023, doi:10.3390/jcm12227069_

Round 1
Reviewer 1 Report (Previous Reviewer 1)
Comments and Suggestions for Authors
The authors have done a great work in simplifying the tables and figures which improved the presentation of the findings and the manuscript in general.
Author Response
Thank you for your comment.
Reviewer 2 Report (New Reviewer)
Comments and Suggestions for Authors
An important article, that is needed and useful not only in the Spanish context but globally. This article can be strengthened with use of Spanish methodolgy, or a section on how Spanish culture affects the study. The below articles will also strengthen this article.
Restubog, S. L. D., Ocampo, A. C. G., & Wang, L. (2020). Taking control amidst the chaos: Emotion regulation during the COVID-19 pandemic. Journal of vocational behavior, 119, 103440.
Martinelli, N., Gil, S., Belletier, C., Chevalère, J., Dezecache, G., Huguet, P., & Droit-Volet, S. (2021). Time and emotion during lockdown and the Covid-19 epidemic: Determinants of our experience of time?. Frontiers in Psychology, 11, 616169.
Author Response
Dear Reviewer, thank you for your recommendations to support our research. Now, you can see that both references were included in the text.
Reviewer 3 Report (New Reviewer)
Comments and Suggestions for Authors
Dear Authors:
I have reviewed the manuscript entitled “Emotional distress caused by the measures taken in assisted reproductive treatments during the COVID-19 confinement in Spain”. This study aims to evaluate the level of anxiety of women whose treatments were either suspended or delayed and to evaluate the different sociodemographic variables associated to the emotional management of this situation. It was conducted a Descriptive cross-sectional study conducted between April and May 2020. It was used a structured survey developed by the Human Reproduction Unit team and the State-Trait Anxiety Inventory (STAI) applied in a Spanish adapted version, in 115 patients. Participants recruitment followed a consecutive random sampling conducted during the confinement months among women included in the assisted reproduction program at the Canarias university hospital. Patients were interviewed on the phone after obtaining their verbal consent to participate and receiving the information regarding objectives and implications of collaborating in the study. The outcomes suggested that Discontinuation of fertility treatments due to confinement restrictions had a negative impact on the mental health of women who were following a process of assisted reproduction treatment, increasing their levels of emotional distress and anxiety.
The thematic is very interesting and the outcomes are especially important in health care provided to women in fertilization programs. I have some suggestions to improve your paper:
1) Page 1 line 36: please review the sentence: “symptoms of emotional de emotional affectation” and in page 9 lines 231-233 “There were no differences in the resto f psychological variables, in line with the results of the study by Haham et al [16]. Whereas in this study the most common emotional reactions to the suspension of treatment were sadness and y anxiety”.
2) In abstract and data instrument description is not clear that was used interviews by phone (I believe it could be improved)?
3) The limitations of study (as interviews by phone) and suggestions to future investigation is needed.
4) Did you use the STROBE checklist to confirm all point of your paper? I suggest attaching the checklist; (https://www.equator-network.org/reporting-guidelines/strobe/)
I have nothing to add and I wish you good luck towards publishing the paper!
Best regards
Author Response
Dear Authors:
I have reviewed the manuscript entitled “Emotional distress caused by the measures taken in assisted reproductive treatments during the COVID-19 confinement in Spain”. This study aims to evaluate the level of anxiety of women whose treatments were either suspended or delayed and to evaluate the different sociodemographic variables associated to the emotional management of this situation. It was conducted a Descriptive cross-sectional study conducted between April and May 2020. It was used a structured survey developed by the Human Reproduction Unit team and the State-Trait Anxiety Inventory (STAI) applied in a Spanish adapted version, in 115 patients. Participants recruitment followed a consecutive random sampling conducted during the confinement months among women included in the assisted reproduction program at the Canarias university hospital. Patients were interviewed on the phone after obtaining their verbal consent to participate and receiving the information regarding objectives and implications of collaborating in the study. The outcomes suggested that Discontinuation of fertility treatments due to confinement restrictions had a negative impact on the mental health of women who were following a process of assisted reproduction treatment, increasing their levels of emotional distress and anxiety.
The thematic is very interesting and the outcomes are especially important in health care provided to women in fertilization programs. I have some suggestions to improve your paper:
1) Page 1 line 36: please review the sentence: “symptoms of emotional de emotional affectation” and in page 9 lines 231-233 “There were no differences in the resto f psychological variables, in line with the results of the study by Haham et al [16]. Whereas in this study the most common emotional reactions to the suspension of treatment were sadness and y anxiety”.
Response: Thank you for these comments that were amended.
2) In abstract and data instrument description is not clear that was used interviews by phone (I believe it could be improved)?
Response: This suggestion was followed and included. Please, see lines 18-19 and 78-79
3) The limitations of study (as interviews by phone) and suggestions to future investigation is needed.
Response: Thank you for the recommendation. This was included 263-264.
4) Did you use the STROBE checklist to confirm all point of your paper? I suggest attaching the checklist; (https://www.equator-network.org/reporting-guidelines/strobe/)
Response: Thank you. This was included.
This manuscript is a resubmission of an earlier submission. The following is a list of the peer review reports and author responses from that submission.
Round 1
Reviewer 1 Report
Comments and Suggestions for Authors
The authors of this work investigated the emotional distress on Spanish people imposed by the measures taken in assisted reproductive treatment during the COVID-19 pandemic.
The introduction contains very long sentences that need to be revised and re-written for example the sentence from line 33 to line 38.
The tables used in presenting the results are crowded, confusing and difficult to follow. Please modify, simplify and present the important points and findings only.
Figures are so crowded and too busy. Please try to present the data in simplified figures.
Nothing is mentioned in regard to the number of cycles the patient had before the one stopped because of COVID. This would reflect the amount of stress and anxiety the patients had before the stop of the last cycle.